# The Value of Left Ventricular Mechanical Dyssynchrony and Scar Burden in the Combined Assessment of Factors Associated with Cardiac Resynchronization Therapy Response in Patients with CRT-D

**DOI:** 10.3390/jcm12062120

**Published:** 2023-03-08

**Authors:** Tariel A. Atabekov, Mikhail S. Khlynin, Anna I. Mishkina, Roman E. Batalov, Svetlana I. Sazonova, Sergey N. Krivolapov, Victor V. Saushkin, Yuliya V. Varlamova, Konstantin V. Zavadovsky, Sergey V. Popov

**Affiliations:** Cardiology Research Institute, Tomsk National Research Medical Center, Russian Academy of Sciences, Kievskaya St., 111a, Tomsk 634012, Russia

**Keywords:** heart failure, left bundle-branch block, mechanical dyssynchrony, response to cardiac resynchronization therapy, gated SPECT myocardial perfusion imaging

## Abstract

Background: Cardiac resynchronization therapy (CRT) improves the outcome in patients with heart failure (HF). However, approximately 30% of patients are nonresponsive to CRT. The aim of this study was to determine the role of the left ventricular (LV) mechanical dyssynchrony (MD) and scar burden as predictors of CRT response. Methods: In this study, we included 56 patients with HF and the left bundle-branch block with QRS duration ≥ 150 ms who underwent CRT-D implantation. In addition to a full examination, myocardial perfusion imaging and gated blood-pool single-photon emission computed tomography were performed. Patients were grouped based on the response to CRT assessed via echocardiography (decrease in LV end-systolic volume ≥15% or/and improvement in the LV ejection fraction ≥5%). Results: In total, 45 patients (80.3%) were responders and 11 (19.7%) were nonresponders to CRT. In multivariate logistic regression, LV anterior-wall standard deviation (adjusted odds ratio (OR) 1.5275; 95% confidence interval (CI) 1.1472–2.0340; *p* = 0.0037), summed rest score (OR 0.7299; 95% CI 0.5627–0.9469; *p* = 0.0178), and HF nonischemic etiology (OR 20.1425; 95% CI 1.2719–318.9961; *p* = 0.0331) were the independent predictors of CRT response. Conclusion: Scar burden and MD assessed using cardiac scintigraphy are associated with response to CRT.

## 1. Introduction

Heart failure (HF) is a rapidly growing public health issue, with an estimated prevalence of more than 37.7 million individuals globally [1]. In the developed world, this disease affects approximately 2.0% of the adult population [2]. In the United States, the total percentage of the population with HF is projected to rise from 2.4% in 2012 to 3.0% in 2030 [3,4]. In Russian Federation, the prevalence of chronic HF (CHF) is 10.2% [5]. The main cause of CHF is coronary heart disease, which accounts for about 70.0%, and the remaining 30.0% are nonischemic heart diseases [6]. More than two decades of research has established the role of cardiac resynchronization therapy (CRT) in medically refractory, mild to severe systolic HF with abnormal QRS duration and morphology [7]. The prolongation of QRS (120 ms or more) occurs in 14.0% to 47.0% of HF patients and the ventricular conduction disturbance, most commonly the left bundle-branch block (LBBB), is present in approximately one-third of HF, leading to the mechanical dyssynchrony (MD) of ventricles [6,8]. Prospective randomized studies of patients with both ischemic HF (IHF) and nonischemic HF (NIHF) have shown that CRT translates into long-term clinical benefits, such as improved quality of life, increased functional capacity, reduction in hospitalization for HF, and overall mortality [9,10,11]. These patients qualified as responders to CRT [12,13]. However, CRT is effective in 70.0% of patients, and the remaining 30.0% do not respond to the device therapy [7,14]. This finding suggests that the existing criteria for patient selection to CRT are not always effective. In this regard, it is necessary to search for additional selection criteria or prognostic markers for CRT response in patients with CHF.

Some studies have shown that the ventricular MD assessment using transthoracic echocardiography (TTE), magnetic resonance imaging (MRI), computed tomography (CT), and single-photon emission computed tomography (SPECT) can play a key role in the prediction of CRT response [15,16,17,18,19]. However, according to other studies, TTE indicators are not reliable CRT response predictors [20]. The application of MRI, despite its high accuracy and information content, is limited by its high cost and complexity of cardiac protocols, and it is an operator-dependent method [21]. By contrast, gated myocardial perfusion imaging (MPI) and gated blood-pool SPECT (gBPS) are simpler, have higher reproducibility, and allow for the detection of the area of ventricular MD and impaired myocardial flow and scarring [22]. The predictive value of MD and scar burden assessed via MPI remains disputable. In some studies, MD is correlated with the positive response to CRT in patients with HF of ischemic and nonischemic etiologies [23,24]. However, another work shows the absence of MD prognostic significance in patients with IHF [25,26]. The good prognostic value of scar burden assessed via cardiac scintigraphy has been demonstrated in several studies [27]. However, papers devoted to the combined assessment of the factors associated with reverse remodeling, which can be used for the improved selection of patients for CRT, are rare [28].

The aim of this study was to identify the significance of MD and scar burden assessment using MPI and gBPS as a combined predictor of CRT response in patients with HF of ischemic and nonischemic etiologies. 

## 2. Materials and Methods

### 2.1. Patient Population and Study Design

Patients with indications for CRT according to the ESC guidelines were included in this clinical, nonrandomized, open, prospective study [29]. The inclusion and exclusion criteria were determined in accordance with the research project “Single-Photon Emission Computed Tomography for Prediction and Evaluation of Cardiac Resynchronization Therapy Efficacy in Chronic Heart Failure Patients” (ClinicalTrials.gov, NCT03667989). Patients who were included in the study met the following criteria: the presence of HF (ischemic or nonischemic etiology), sinus rhythm, permanent LBBB with QRS duration ≥150 ms, New York Heart Association (NYHA) functional class (FC) of HF II-III, left ventricular ejection fraction (LVEF) ≤35%, and optimal medical therapy for at least 3 months. Individuals with NYHA FC of HF 0, I, IV, decompensated HF, recent myocardial infarction (less than 3 months), recent revascularization (within last 3 months), other cardiac intervention, acute HF decompensation, right bundle-branch block, previously implanted pacemaker or cardioverter–defibrillator, severe comorbidity, cognitive impairment, or with indications of revascularization and heart transplantation, as well as patients under 18 years of age, were excluded. HF etiology was considered ischemic in the presence of significant coronary artery disease (≥50% stenosis in one or more of the major coronary arteries) and/or a history of myocardial infarction or prior revascularization. 

All patients underwent full physical examination (6 min walk distance test (6MWDT), electrocardiography (ECG), transthoracic echocardiography (TTE), Holter ECG monitoring, coronary angiography, and blood analyses), MPI with 99mTc–methoxy isobutyl isonitrile (99mTc–MIBI) and gBPS before device implantation. In all cases, CRT devices with the defibrillation function (CRT-D) were implanted, according to the ESC guidelines [29]. All patients received the basic therapy in accordance with the present guidelines. The follow-up was performed 6 months after CRT-D implantation.

### 2.2. Consent

The study was carried out in accordance with the principles of the Helsinki Declaration and with the standards of good clinical practice. The study protocol was approved by the local ethics committee. All the participants received written informed consent prior to the study inclusion. Ethical approval by the hospital review committee and patient consent according to the institutional guidelines were obtained. 

### 2.3. Minute Walk Distance Test

The assessment of HF FC was performed in accordance with the NYHA criteria, using 6MWDT before and 6 months after CRT-D implantation. For the analysis, the walking distance in meters and NYHA FC were used, and the following ranking was used:-More than 551 m—the patient has no signs of HF;-A distance of 426–550 m—I FC of HF;-A distance of 301–425 m—II FC of HF;-A distance of 151–300 m—III FC of HF;-Less than 150 m—IV FC of HF.

### 2.4. TTE Acquisition and Analysis

TTE with the intracardiac hemodynamic parameter assessment was performed using Philips HD15 PureWave (the Netherlands) ultrasound machine before and after 6 months of CRT-D implantation. The examination was carried out from standard positions with the determination of the left atrium (LA) and right ventricle (RV) sizes, interventricular septum (IVS) and LV posterior-wall (LVPW) thickness, LV end-systolic dimension (LVESD), LV end-diastolic dimension (LVEDD), LV end-systolic volume (LVESV), LV end-diastolic volume (LVEDV), the index of myocardial mass (IMM), LVEF, right ventricular systolic pressure (RVSP), stroke volume (SV), LV end-systolic index (LVESI), LV end-diastolic index (LVEDI), the left atrial index (LAI), and the right atrial index (RAI). The mitral, tricuspid, and aortic valve functions, as well as the right and left ventricular contractility, were assessed. 

### 2.5. Scintigraphic Data Acquisition

The scintigraphy examination was performed using CZT SPECT/CT (GE Discovery 570C, GE Healthcare, Haifa, Israel) with low-energy multi-pinhole collimators and 19 stationary detectors [30]. Each detector contained 32 × 32 pixelated (2.46 × 2.46 mm) CZT elements. The energy window was symmetrically centered to ± 20% of the 140 keV Tc-99m photopeak. The images were reconstructed on the dedicated workstation (Xeleris 4.0; GE Healthcare, Haifa, Israel).

The time interval between MPI and gBPS examinations ranged from 1 to 2 days. MPI was performed first in all patients. 

### 2.6. MPI Acquisition and Analysis

MPI was performed according to the standard ECG-gated (16 frames/cardiac cycle) rest protocol [30] approximately 60 min after the administration of 138–357 MBq of 99mTc–MIBI. All patients were imaged in a supine position with arms placed over their heads for an acquisition time of 7 min. A low-dose CT scan (120 kV, 20 mA) was performed for the attenuation correction. All images were reconstructed using the iterative reconstruction (60 iterations; Green OSL α 0.7; Green OSL β 0.3) and Butterworth post-processing filter (frequency 0.37; order 7) in a 70 × 70-pixel matrix with 50 slices. MPI data were processed using Corridor 4DM (University of Michigan, Ann Arbor, MI, USA) software. The reconstructed MPI included the standard cardiac short and vertical and horizontal long axes and the 17-segment bull’s eye map. Each of the 17 segments was scored based on a semiquantitative 5-point scoring system (from 0—normal uptake to 4—no radiotracer uptake) [31]. The summed rest score (SRS) of all segments was quantified. Scar burden was defined as all segments with abnormal radiotracer uptake at rest.

### 2.7. gBPS Acquisition and Analysis

gBPS was performed following in vivo labeling patient’s red blood cells with a 99mTc–pertechnetate dose of 555–720 MBq [32]. The data were acquired with ECG gating (16 frames/cardiac cycle). The patients were imaged in a supine position with arms placed over their heads for an acquisition time of 10 min. No attenuation correction was used. Images were reconstructed using iterative reconstruction (60 iterations; Green OSL α 0.7; Green OSL β 0.3) and a Butterworth post-processing filter (frequency 0.52; order 5) in a 70 × 70-pixel matrix with 57 slices. The image quantification and phase analysis were obtained with the Quantitative Blood-Pool SPECT 2009.0 (Cedars-Sinai Medical Center, Los Angeles, CA, USA) software, which allowed for an evaluation of the functional variables of LV and RV. Ventricular contours were adjusted manually when required. The following parameters for both ventricles were determined: the peak emptying rate (PER, expressed as EDV/s), the peak filling rate (PFR, EDV/s), and the second peak filling rate (PFR2, EDV/s).

The severity of intra- and interventricular dyssynchrony was evaluated using the Fourier transform method. Global MD indices were evaluated for ventricles’ phase standard deviation (SD), histogram bandwidth (HBW), and phase entropy (PE). Interventricular dyssynchrony (IVD) was calculated according to the histogram peak of LV and RV. Moreover, the following regional MD indices were assessed: SD and E. The regional analysis of LV MD was based on the assessment of the free wall (FW), anterior wall (AW), lateral wall (LW), inferior wall (IW), and septal wall (SW) of LV. The mean effective radiation dose for the entire study protocol was 7.48 ±1 mSv (range 5.1–10.3 mSv) per patient.

### 2.8. CRT-D Implantation and Programming

The active-fixation atrial (AL) and defibrillation leads (DLs), as well as the passive-fixation LV lead, were positioned under fluoroscopic guidance using a transvenous approach. DL was implanted into the right ventricular apex or interventricular septum. Lead positions were confirmed via fluoroscopy in the posteroanterior (PA) and left anterior oblique (LAO) view and the intraoperative threshold testing. The capture threshold, sensing amplitude, and impedance measurements of the leads were performed using the pacing system analyzer (Medtronic, Minneapolis, MN, USA) with sterile crocodile clip cables. 

The implantation of the LV pacing lead was performed by cannulating one of the tributaries of the coronary sinus using the delivery system. The venogram was performed in AP and LAO projections. Access to the target branch within the coronary sinus and subsequent advancement of the LV lead was performed using an “over-the-wire” technique. In case of troubles with the target vein branch cannulation, we used the “interventional” technique with inner catheters (subselector). The intraoperative threshold testing was performed for checking the optimal pacing threshold and phrenic nerve stimulation after LV lead was positioned to the target vein.

CRT-D programming was carried out in accordance with international standards [33]. In every CRT-D device, the monitoring zone was programmed with a heart rate of 140–170 beats per minute (bpm) for more than 50 consecutive cycles without antitachycardia pacing (ATP) and shock therapy. The ventricular tachycardia (VT) zone was programmed for 170–200 bpm with 30 cycles and with ATP (≥1 burst pacing and ≥1 ramp pacing) and shock therapy (first shock with the submaximal shock discharge). The ventricular fibrillation (VF) zone was programmed for ≥201 bpm with 12 cycles and with ATP during CRT-D charging and the maximum shock discharge. The atrioventricular (AV) delay interval was programmed 20–40 ms less than the native AV delay. Biventricular pacing was switched on in all patients, and the V–V interval (left ventricle → right ventricle) was optimized using the surface ECG to obtain the narrowest QRS complex. 

### 2.9. Clinical Follow-Up, CRT-D Data Acquisition, and Analysis

The follow-up information was acquired for 56 of the 64 patients who were included in the prospective analysis. The information about arrhythmic events was determined from CRT-D interrogation reports. The arrhythmic events (VT, VF, atrial fibrillation, and the appropriate and inappropriate ICD therapy) and CRT-D lead system parameters were evaluated at follow-up. The flowchart of the study is shown in Figure 1. The primary endpoint was a decrease in LVESV ≥15% or/and an improvement in LVEF ≥5% [26,34].

### 2.10. Statistical Analysis 

Statistical analysis was performed using the software package Statistica 10.0, StatSoft (USA). The Shapiro–Wilk test was used to assess the normality of the distribution of the trait. For variables with the normal distribution, the mean value (M) and standard deviation (SD) were calculated, while for others, the median (Me) with an interquartile range [Q₁, Q₃] were calculated. The nonparametric Mann–Whitney test for independent samples and the Wilcoxon test for dependent samples were used. The nonparametric Spearman analysis was used to assess the correlations between the pairs of quantitative features. Efficacy analysis was performed using logistic regression analysis. The forward-stepwise logistic regression analysis was used to evaluate the independent predictors of CRT response. Receiver operating characteristic (ROC) analysis was used to determine the diagnostic efficiency of the method using MedCalc statistical software package. We considered significant *p* values <0.05.

## 3. Results

### 3.1. Patients’ Baseline Clinical and Follow-Up Characteristics

A total of 56 (100.0%) patients who underwent MPI, gBPS, CRT-D implantation, and 6-month follow-up were included in this study. The first group consisted of 45 (80.3%) individuals with a response to CRT (RESP), and the second group comprised 11 (19.7%) patients without response (non-RESP). The baseline demographics and clinical characteristics of the included patients are shown in Table 1. Considering all patients, the mean age was 57.0 ± 11.5 years, and 35 (62.5%) patients were males. The baseline 6MWDT, HF class NYHA, QRS duration, TTE parameters, arrhythmias before CRT-D implantation, LV lead position, HF etiology, comorbid pathologies, and medical therapy records between the groups are also shown in Table 1. There were no significant differences between the groups in baseline demographics and clinical characteristics.

Both groups were comparable in terms of pre-CRT-D implantation scintigraphic parameters, except SRS, RV PFR, and MD indicators, assessed via MPI and gBPS. In RESP patients, the right ventricular peak filling rate (RV PFR) (*p* = 0.005), the left ventricular anterior-wall entropy (LV AW_E) (*p* = 0.001), the left ventricular anterior-wall standard deviation (LV AW_SD) (*p* = 0.0001), the right ventricular free-wall standard deviation (RV FW_SD) (*p* = 0.011), the left ventricular entropy (LV_E) (*p* = 0.033), and IVD (*p* = 0.022) were significantly higher, and SRS (*p* = 0.018) and RV PFR2 (*p* = 0.028) were significantly lower than in non-RESP subjects. A detailed comparison of pre-CRT-D implantation scintigraphic parameters is presented in Table 2. 

Both groups were comparable in terms of TTE parameters before CRT-D implantation. In RESP patients 6 months after CRT-D implantation, LVESD (*p* = 0.015), LVEDD (*p* = 0.04), LVESV (*p* = 0.005), LVESI (*p* = 0.019), and LVSI (*p* = 0.02) were significantly lower, and LVEF (*p* < 0.001) was significantly higher than in non-RESP patients. The QRS duration 6 months after CRT-D implantation did not differ significantly between the groups. In RESP patients, QRS duration was 140.0 ms [130.0; 140.0], and non-RESP was 140.0 ms [130.0; 140.0] (*p* = 0.243).

Both groups were comparable in terms of the number of I (*p* = 0.215) and II (*p* = 0.332) HF class NYHA patients after 6 months of CRT-D implantation. The quantity of III HF class NYHA patients in the non-RESP (*n* = 7 (63.6%)) group was significantly higher than in the RESP (*n* = 9 (20.0%)) group (*p* = 0.026). Additionally, 6MWDT in the RESP (382.4 ± 88.1 m) group was significantly higher than in the non-RESP (288.1 ± 73.1 m) group (*p* = 0.003) after 6 months of CRT-D implantation. 

### 3.2. Events According to the CRT-D Interrogation Data

During 6 months of follow-up, VT events were registered in seven (12.5%) patients from both groups. In the first group (RESP patients), VT was registered in four (7.1%) patients: three unsustained VT with spontaneous termination and one sustained VT, terminated using ATP therapy. In the second group (non-RESP patients), VT was registered in three (5.4%) cases (*p* = 0.353). There were only three unsustained VT with spontaneous termination (*p* = 0.297). There were no lead dysfunction, dislocation, and inappropriate ICD therapy in either group. In RESP patients 6 months after device implantation, according to the CRT-D interrogation data, the paced atrioventricular delay was 148.7 ± 14.5 ms, the sensed atrioventricular delay was 110.0 ± 16.1 ms, the interventricular delay (first LV) was 25.0 ms [15.0; 40.0], and the time of biventricular pacing was 98.1 ± 1.9%. In non-RESP individuals 6 months after CRT-D implantation, the paced atrioventricular delay was 153.6 ± 18.0 ms (*p* = 0.464), the sensed atrioventricular delay was 116.8 ± 18.2 ms (*p* = 0.215), the interventricular delay (first LV) was 20.0 ms [10.0; 40.0] (*p* = 0.885), and the time of biventricular pacing was 98.7 ± 0.4% (*p* = 0.433).

### 3.3. CRT Response Predictors

According to the ROC analysis, the SRS and MD indicators (RV PFR, LV AW_E, LV AW_SD, RV FW_SD, LV_E, IVD, and RV PFR 2) were statistically significant predictors of CRT response (Table 3). 

The multivariate logistic regression analysis included the following indicators: QRS duration, gender, HF etiology, LV lead position, 6 min walk distance test, LVESV, LVEF, SRS, and MD. This analysis showed that only myocardial perfusion defects and MD indicators assessed via MPI and gBPS such as LV AW_SD (OR 1.5275; 95% CI 1.1472–2.0340; *p* = 0.0037) and SRS (OR 0.7299; 95% CI 0.5627–0.9469; *p* = 0.0178) and also HF with a nonischemic etiology (OR 20.1425; 95% CI 1.2719–318.9961; *p* = 0.0331) were the independent predictors of CRT response.

The CRT response prediction model was performed considering the combination of LV AW_SD, SRS, and the presence of nonischemic HF etiology. When ROC analysis was performed with the prediction model, the AUC was 0.949, Sen = 95.56, Spe = 81.82, and *p* < 0.001 (Figure 2).

The pairwise comparison of CRT response probability indicators revealed that the ROC curve with SRS (*p* = 0.001), HF nonischemic etiology (*p* < 0.001), the mechanical dyssynchrony indicators assessed via MPS and gBPS (IVD (*p* = 0.01), LVE (*p* < 0.001), RV PFR 2 (*p* = 0.003), RV PFR (*p* = 0.005), LV AW_E (*p* = 0.037), and RV FW_SD (*p* = 0.014)) ROC curves showed significant differences, with the exception of LV AW_SD (*p* = 0.089) (Table 4). 

## 4. Discussion

In the present study, it was shown that mechanical dyssynchrony (left ventricular anterior-wall standard deviation) and the scar burden assessed via cardiac scintigraphy in patients with HF of different etiologies may have prognostic value. Thus, HF with a nonischemic etiology, a low value of SRS, and a high value of LV AW_SD are favorable predictive indicators of CRT response. This may be due to the smaller size of the scar and the involvement of the anterior LV wall in mechanical dyssynchrony.

The nonischemic origin of HF was previously reported to predict reverse remodeling in patients with more advanced HF symptoms. According to Ypenburg et al., CRT responders and patients with super responses more frequently had a nonischemic etiology of HF [35]. In a study by Verhaert et al., it was shown that the female gender and the nonischemic etiology of HF were associated with a much greater initial response to CRT [36]. Another study by Said et al. found that women showed a greater echocardiographic response to CRT at 6 months of follow-up [37]. However, after the adjustment for body surface area and ischemic etiology, no differences were found in LV function measures or survival, suggesting that having a nonischemic etiology is responsible for greater response rates in women treated with CRT [37]. In our study, the number of patients with ischemic HF and nonischemic HF did not significantly differ in the groups of responders and nonresponders (*p* = 0.124). However, multivariate logistic regression showed that the nonischemic etiology of HF is an independent predictor of CRT response (95% CI 1.2719–318.9961; *p* = 0.0331). Furthermore, the subanalyses of multiple prospective randomized studies, including the Multicenter InSync Randomized Clinical Evaluation (MIRACLE), Cardiac Resynchronization—Heart Failure (CARE-HF), and Multicenter Automatic Defibrillator Implantation with Cardiac Resynchronization Therapy (MADIT-CRT), have confirmed the finding of greater reverse remodeling in HF with a nonischemic etiology [28,38].

The clinical significance of the scar burden assessed with MPI was demonstrated in several studies. Thus, in a study by Adelstein E. et al., it was found that among ischemic cardiomyopathy patients, lesser scar burden evaluated via SPECT MPI (SRS ˂ 27) was associated with more favorable survival and reverse remodeling following CRT, with outcomes similar to nonischemic cardiomyopathy patients [27]. High scar burden (SRS ≥ 27) was associated with the lack of LV functional improvement, the absence of reverse remodeling, and worse survival [27]. In our study, patients with HF of different etiologies were included, and a comparison of the groups with ischemic and nonischemic etiologies was not performed. In patients with HF, myocardial perfusion impairment can vary widely and may be associated with the presence of myocardial fibrosis due to myocardial infarction in ischemic HF patients, or the presence of interstitial fibrosis, which may play a crucial role in the process of myocardial remodeling [39]. Recent studies revealed that myocardial fibrosis is an independent predictor of mortality and morbidity in patients with dilated cardiomyopathy undergoing CRT [40]. Additionally, in our study, which included mixed samples of patients with ischemic and nonischemic etiologies, the SRS assessed using MPI ≤ 7.0 was an independent significant predictor for CRT patient selection (95% CI 0.5627–0.9469; *p* = 0.0178).

The predictive value of LV mechanical dyssynchrony for CRT patient selection measured via MPI and gBPS was widely studied. In a study with 142 CRT patients, LV mechanical dyssynchrony parameters such as the systolic histogram bandwidth (95% CI 0.98–1.00, *p* = 0.041), the diastolic-phase standard deviation (95% CI 0.94–1.00, *p* = 0.041), and the diastolic histogram bandwidth (95% CI 0.98–1.00, *p* = 0.028) were significant independent predictors of CRT response only for nonischemic HF patients [25]. For ischemic HF individuals, none of the LV mechanical dyssynchrony parameters were statistically significant [25]. In a study by Henneman M. et al. with 42 CRT patients, the ROC analysis showed that the optimal cut-off values for the phase standard deviation and histogram bandwidth were 43° (sensitivity and specificity of 74%) and 135° (sensitivity and specificity of 70%), respectively [41]. In a study with 324 consecutive individuals with nonischemic HF CRT patients, it was shown that the systolic-phase standard deviation, adjusted to age, hypertension, diabetes, aspirin, beta-blockers, diuretics, QRS, and LVEF, was an independent predictor of all-cause mortality (HR 1.97, 95% CI 1.06–3.66, *p* = 0.033) [42]. In our study, the responders and nonresponders did not differ in terms of such indicators of global LV mechanical dyssynchrony as the phase standard deviation and histogram bandwidth, and these indicators had no prognostic value, but the LV phase entropy was higher in the responders’ sample. However, the novelty of our study was the use of gBPS for assessing the regional mechanical dyssynchrony indicators separately for the septum, anterior, posterior, and lateral walls of LV. These indicators (LV AW_E, LV AW_SD, and RV FW_SD) with IVD and contractile function indicators (RV PFR and RV PFR 2) were statistically significant predictors of CRT response. The multivariate logistic regression showed that LV AW_SD (OR 1.5275; 95% CI 1.1472–2.0340; *p* = 0.0037) was the independent predictor of CRT response. This may indicate that the assessment of regional myocardial dyssynchrony may provide additional information for successful resynchronization therapy. However, disagreements with previous studies underline the importance of these findings and the need for future large-scale studies.

There are few publications about the combined assessment of the factors associated with reverse remodeling, which can be used for the improved selection of patients for CRT. The significance of the CRT response score was shown in a major randomized trial MADIT-CRT (*n* = 1761) [28]. This score included seven factors associated with the echocardiographic response to CRT-D and made up the response score (female sex, nonischemic HF, LBBB, QRS duration ≥150 ms, prior hospitalization for HF, left ventricular end-diastolic volume ≥125 mL/m^2^ and left atrial volume ˂40 mL/m^2^). The multivariate analysis showed a 13% (*p* ˂ 0.001) increase in the clinical benefit of CRT-D per one-point increment in the response score (range, 0–14) and a significant direct correlation between the risk reduction associated with CRT-D and the response score quartiles: Patients in the first quartile did not reveal a significant reduction in the risk of HF or death with CRT-D (hazard ratio = 0.87; *p* = 0.52); patients in the second and third quartiles had 33% (*p* = 0.04) and 36% (*p* = 0.03) risk reductions, respectively; and patients in the upper quartile experienced a 69% (*p* ˂ 0.001) risk reduction (*p* = 0.005). In our study, the multivariate logistic regression with inclusion factors such as QRS duration, gender, HF etiology, LV lead position, 6 min walk distance test, LVESV, LVEF, SRS, and mechanical dyssynchrony indicators assessed via gBPS showed that LV AW_SD, SRS, and HF with a nonischemic etiology were the independent predictors of CRT response (95% CI 0.856–0.990; AUC = 0.949; Sen = 95.56; Spe = 81.82; *p* < 0.001).

According to a study by Forleo et al., quadripolar leads allowed for nonapical pacing with acceptable electrical parameters in the majority of CRT recipients, although differences were found among the currently available devices [43]. In our study, quadripolar and bipolar LV leads from various manufacturers were implanted. A detailed analysis of lead characteristics between different companies was not performed. However, a comparison between the groups of responders and nonresponders to CRT in terms of the number of implanted quadripolar and bipolar LV leads was made. The comparative analysis between the groups did not reveal significant differences (*p* = 0.392).

Thus, the predictive model with the combination of LV AW_SD, SRS, and the presence of nonischemic HF etiology has a high prognostic value and can be used as an additional CRT response predictor in patients with HF of different etiologies.

### Study Limitations

The limitations of this study included its relatively small sample size and short follow-up period; the left ventricular mechanical dyssynchrony indicators were not determined in terms of their dynamics, and this was a nonrandomized, single-center study. 

## 5. Conclusions

In patients with HF of ischemic and nonischemic etiologies, the left ventricular mechanical dyssynchrony assessed via gBPS may be useful in identifying responders to CRT. In these patients, a combined assessment of the factors associated with reverse remodeling (HF with a nonischemic etiology, LV AW_SD, and SRS) can be used for the improved selection of patients for CRT. 

## Figures and Tables

**Figure 1 jcm-12-02120-f001:**
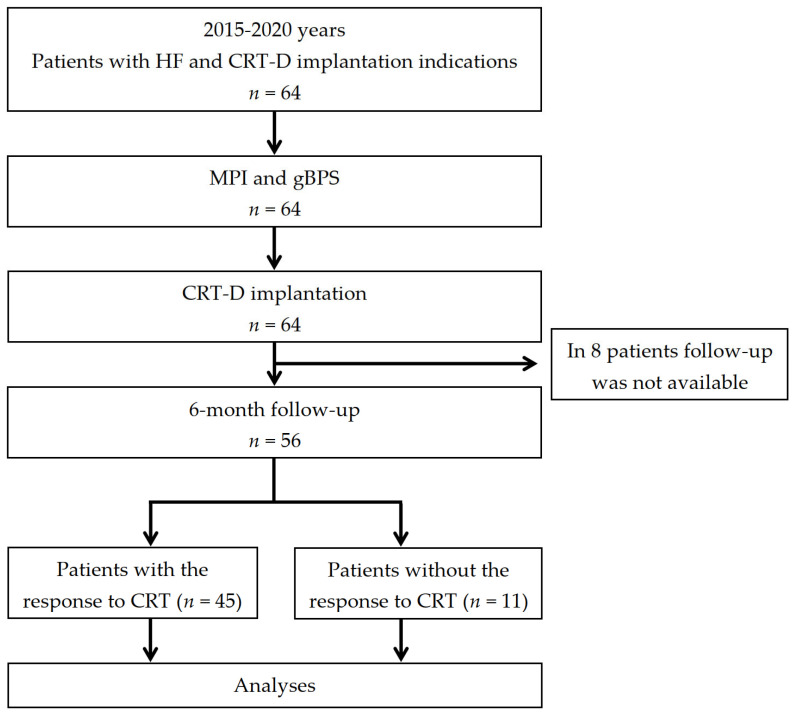
Study flowchart: HF—heart failure; MPI—myocardial perfusion imaging; gBPS—gated blood-pool SPECT; CRT—cardiac resynchronization therapy; CRT-D—CRT devices with the defibrillation function.

**Figure 2 jcm-12-02120-f002:**
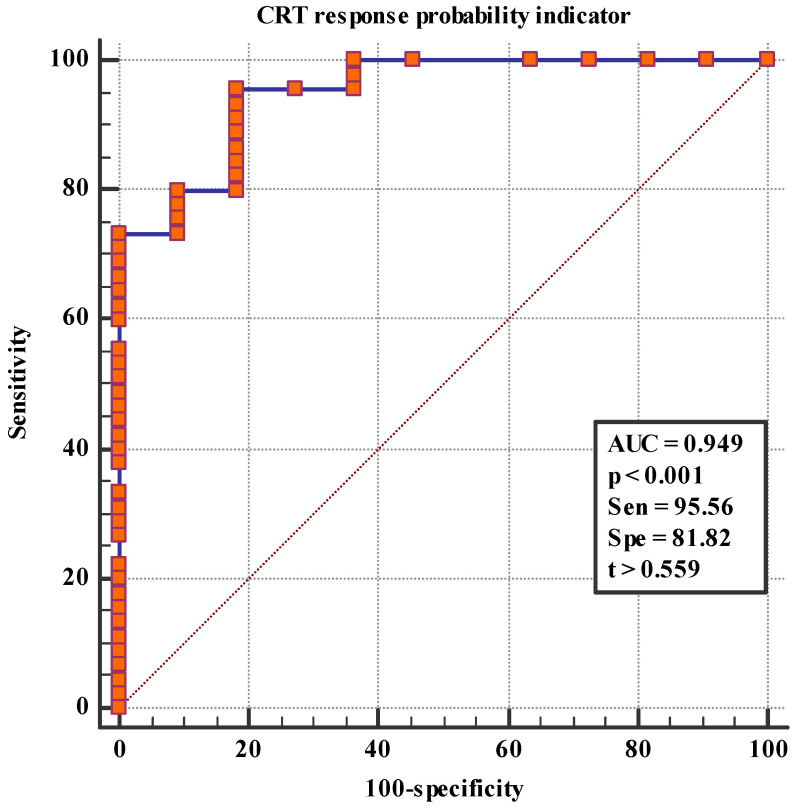
CRT response probability indicator, calculated according to the predictive model. AUC—area under the curve, Sen—sensitivity, Spe—specificity, t—threshold value.

**Table 1 jcm-12-02120-t001:** Baseline demographics and clinical characteristics of enrolled patients.

Demographic and Clinical Characteristics	Total (*n* = 56)	RESP (*n* = 45)	Non-RESP (*n* = 11)	P_2–3_
	1	2	3	
Age, year, mean ± SD	57.0 ± 11.5	56.7 ± 11.8	58.4 ± 10.8	0.680
Male gender, *n* (%)	35 (62.5)	28 (62.2)	7 (63.6)	0.950
Ischemic heart failure, *n* (%)	22 (39.3)	15 (33.4)	7 (63.6)	0.124
Nonischemic heart failure, *n* (%)	34 (60.7)	30 (66.6)	4 (36.4)	0.124
6 min walk distance test, m, mean ± SD	284.5 ± 66.3	288.3 ± 64.8	269.0 ± 73.1	0.312
Heart failure:				
II functional class NYHA, *n* (%)	23 (41.1)	20 (44.4)	3 (27.3)	0.386
III functional class NYHA, *n* (%)	33 (58.9)	25 (55.6)	8 (72.7)	0.386
QRS duration, ms, Me [Q1; Q3]	165.0 [160.0; 180.0]	165.0 [160.0; 180.0]	165.0 [155.0; 185.0]	0.657
LVEF, % [Q1; Q3]	28.0 [22.0; 31.0]	28.0 [21.0; 31.0]	28.0 [25.0; 32.0]	0.502
LVESV, ml [Q1; Q3]	171.0 [131.5; 217.5]	169.0 [133.0; 210.0]	208.0 [121.0; 232.0]	0.598
History of sustained VT, *n* (%)	5 (8.9)	3 (6.7)	2 (18.2)	0.563
History of ventricular fibrillation, *n* (%)	1 (1.8)	1 (2.2)	0 (0.0)	0.917
Comorbidities:				
Hypertension, *n* (%)	22 (39.3)	15 (33.4)	7 (63.6)	0.124
Left ventricular hypertrophy, *n* (%)	45 (80.3)	35 (77.8)	10 (90.9)	0.509
Diabetes mellitus, *n* (%)	8 (14.3)	6 (13.3)	2 (18.2)	0.812
Body mass index, kg/m2, mean ± SD	28.9 ± 5.0	28.7 ± 5.3	29.7 ± 3.8	0.509
Dyslipidemia, *n* (%)	23 (41.0)	18 (40.0)	5 (45.4)	0.788
GFR, ml/min, mean ± SD	72.2 ± 21.2	72.3 ± 22.7	71.8 ± 13.7	0.804
Therapy:				
Beta-blockers, *n* (%)	53 (94.6)	43 (95.5)	10 (90.9)	0.820
Loop diuretics, *n* (%)	44 (78.6)	33 (73.3)	11 (100.0)	0.176
Potassium-sparing diuretics, *n* (%)	43 (76.8)	36 (80.0)	7 (63.6)	0.409
ACEI, *n* (%)	33 (58.9)	26 (57.8)	7 (63.6)	0.772
Antiplatelet agents, *n* (%)	33 (58.9)	24 (53.3)	9 (81.8)	0.148
Statins, *n* (%)	31 (55.3)	25 (55.6)	6 (54.5)	0.967
Amiodarone, *n* (%)	20 (35.7)	16 (35.5)	4 (36.4)	0.975
Angiotensin II receptor blocker, *n* (%)	19 (33.9)	15 (33.4)	4 (36.4)	0.914
LV lead position:				
Lateral vein, *n* (%)	18 (32.1)	16 (35.5)	2 (18.2)	0.380
Posterolateral vein, *n* (%)	20 (35.7)	16 (35.5)	4 (36.4)	0.975
Anterolateral vein, *n* (%)	10 (17.8)	8 (17.7)	2 (18.2)	0.991
Posterior vein, *n* (%)	8 (14.2)	5 (11.1)	3 (27.3)	0.415
Pacing QRS duration, ms, Me [Q1; Q3]	140.0 [130.0; 140.0]	140.0 [130.0; 140.0]	140.0 [130.0; 140.0]	0.243
Quadripolar LV lead, *n* (%)	28 (50.0)	24 (53.3)	4 (36.3)	0.392
Bipolar LV lead, *n* (%)	28 (50.0)	21 (46.7)	7 (63.7)	0.392
Paced AV delay, ms, Me [Q1; Q3]	150.0 [140.0; 150.0]	150.0 [140.0; 150.0]	150.0 [150.0; 150.0]	0.464
Sensed AV delay, ms, Me [Q1; Q3]	120.0 [100.0; 120.0]	120.0 [100.0; 120.0]	120.0 [100.0; 125.0]	0.215
Interventricular delay, ms, Me [Q1; Q3]	22.5 [12.5; 40.0]	25.0 [15.0; 40.0]	20.0 [10.0; 40.0]	0.885

Values are mean ± SD and Me [Q1; Q3] for continuous variables and *n* (%) for categorical variables. ACEI—angiotensin-converting enzyme inhibitors, GFR—glomerular filtration rate, non-RESP—patients without response to cardiac resynchronization therapy, RESP—patients with response to cardiac resynchronization therapy, VT—ventricular tachycardia, NYHA—New York Heart Association, LV—left ventricular, LVEF—left ventricular ejection fraction, LVESV—left ventricular end-systolic volume, AV—atrioventricular.

**Table 2 jcm-12-02120-t002:** Scintigraphic characteristics before CRT-D implantation.

Scintigraphic Parameters	Total (*n* = 56)	RESP (*n* = 45)	Non-RESP (*n* = 11)	P_2–3_
	1	2	3	
Gated Blood-Pool SPECT:				
IVD, ms	67.1 [38.1; 102.4]	71.8 [42.0; 112.7]	39.2 [9.0; 76.7]	0.022
LV HBW, °	203.0 [192.0; 222.0]	203.0 [186.0; 222.0]	216.0 [192.0; 234.0]	0.190
LV PE, %	72.0 [62.0; 73.0]	72.0 [62.0; 73.0]	64.0 [59.0; 66.0]	0.033
RV HBW, °	120.0 [99.0; 198.0]	120.0 [96.0; 186.0]	198.0 [198.0; 204.0]	0.053
RV PE, %	62.0 [59.5; 67.0]	62.0 [60.0; 67.0]	63.0 [59.0; 67.0]	0.375
RV FW_SD, °	28.0 [16.0; 36.5]	28.0 [20.0; 44.0]	16.0 [13.0; 26.0]	0.011
LV S_SD, °	35.0 [23.0; 40.5]	35.0 [25.0; 42.0]	23.0 [19.0; 32.0]	0.105
LV S_E, %	65.0 [56.0; 72.5]	65.0 [59.0; 73.0]	56.0 [50.0; 58.0]	0.061
LV AW_SD, °	25.0 [11.5; 28.0]	25.0 [17.0; 29.0]	10.0 [10.0; 12.0]	<0.001
LV AW_E, %	50.0 [36.0; 57.5]	50.0 [45.0; 61.0]	36.0 [33.0; 37.0]	0.001
LV LW_SD, °	12.0 [9.0; 16.0]	12.0 [9.0; 15.0]	16.0 [9.0; 20.0]	0.109
LV LW_E, %	36.0 [32.0; 47.0]	36.0 [31.0; 47.0]	37.0 [33.0; 52.0]	0.327
LV IW_SD, °	28.0 [26.0; 37.0]	28.0 [26.0; 37.0]	27.0 [25.0; 37.0]	0.312
LV IW_E, %	59.0 [55.5; 61.50]	59.0 [56.0; 61.0]	60.0 [54.0; 62.0]	0.470
LV PER	−0.81 [−1.18; −0.53]	−0.78 [−1.13; −0.44]	−1.01 [−1.38; −0.57]	0.364
LV PFR	0.745 [0.53; 1.045]	0.7 [0.53; 1.00]	0.94 [0.6; 1.27]	0.154
LV PFR2	0.63 [0.535; 0.64]	0.63 [0.48; 0.63]	0.64 [0.58; 0.66]	0.092
RV PER	−1.66 [−2.39; −0.81]	−1.55 [−2.41; −0.6]	−1.72 [−2.17; −1.22]	0.672
RV PFR	1.38 [1.01; 1.71]	1.51 [1.08; 1.77]	1.01 [0.92; 1.08]	0.005
RV PFR2	1.31 [1.23; 1.63]	1.31 [1.22; 1.57]	1.63 [1.39; 1.63]	0.028
Myocardial perfusion imaging:				
SRS, %	7.5 [4.0; 14.0]	6.0 [3.0; 12.0]	13.0 [9.0; 16.0]	0.018

Values are expressed as Me [Q1; Q3]. IVD—interventricular dyssynchrony, LV HBW—left ventricular histogram bandwidth, LV PE—left ventricular phase entropy, RV HBW—right ventricular histogram bandwidth, RV PE—right ventricular phase entropy, RV FW_SD—right ventricular free-wall standard deviation, LV S_SD—left ventricular septal-wall standard deviation, LV S_E—left ventricular septal-wall entropy, LV AW_SD—left ventricular anterior-wall standard deviation, LV AW_E—left ventricular anterior-wall entropy, LV LW_SD—left ventricular lateral-wall standard deviation, LV LW_E—left ventricular lateral-wall entropy, LV IW_SD—left ventricular inferior-wall standard deviation, LV IW_E—left ventricular inferior-wall entropy, LV PER—left ventricular peak emptying rate, LV PFR—left ventricular peak filling rate, LV PFR2—left ventricular peak filling rate on the second peak, RV PER—right ventricular peak emptying rate, RV PFR—right ventricular peak filling rate.

**Table 3 jcm-12-02120-t003:** The ROC analysis results.

Parameters	95% CI	Cut-off	AUC	Sensitivity	Specificity	*p*
SRS	0.596–0.841	≤7.0%	0.731	60.00	90.91	0.001
RV PFR	0.639–0.872	˃1.15	0.771	66.67	90.91	˂0.001
LV AW_E	0.689–0.906	˃39.0%	0.815	80.00	90.91	˂0.001
LV AW_SD	0.757–0.947	˃13.0°	0.873	84.44	90.91	˂0.001
RV FW_SD	0.616–0.856	˃27.0°	0.749	64.44	90.91	0.004
LV_E	0.572–0.823	˃68.0%	0.709	68.89	90.91	0.006
IVD	0.590–0.836	˃91.8	0.725	40.00	100.00	0.005
RV PFR 2	0.580–0.829	≤1.38	0.716	73.33	81.82	0.013

Notes: 95% CI—95% confidence interval, AUC—area under the curve, IVD—interventricular dyssynchrony, LV AW_E—left ventricular anterior-wall entropy, LV AW_SD—left ventricular anterior-wall standard deviation, LV_E—left ventricular entropy, RV FW_SD—right ventricular free-wall standard deviation, RV PFR—right ventricular peak filling rate, RV PFR 2—right ventricular peak filling rate on the second peak, SRS—summed rest score.

**Table 4 jcm-12-02120-t004:** The ROC curves’ pairwise comparison.

	AD	MSE	95% CI	*p*
CRT RPI~NIHF	0.298	0.0811	0.139–0.457	<0.001
CRT RPI~SRS	0.218	0.0670	0.0869–0.349	0.001
CRT RPI~LV AW_SD	0.0768	0.0453	−0.0119–0.165	0.089
CRT RPI~IVD	0.224	0.0881	0.0516–0.397	0.010
CRT RPI~LVE	0.240	0.0623	0.118–0.362	<0.001
CRT RPI~RV PFR 2	0.233	0.0804	0.0757–0.391	0.003
CRT RPI~RV PFR	0.179	0.0638	0.0538–0.304	0.005
CRT RPI~LV AW_E	0.134	0.0646	0.00765–0.261	0.037
CRT RPI~RV FW_SD	0.200	0.0817	0.0398–0.360	0.014

Notes: 95% CI—95% confidence interval, AD—area difference, CRT RPI—cardiac resynchronization therapy response probability indicator, IVD—interventricular dyssynchrony, LV AW_E—left ventricular anterior-wall entropy, LV AW_SD—left ventricular anterior-wall standard deviation, LV_E—left ventricular entropy, RV FW_SD—right ventricular free-wall standard deviation, RV PFR—right ventricular peak filling rate, RV PFR 2—right ventricular peak filling rate on the second peak, NIHF—nonischemic heart failure, SRS—summed rest score, MSE—mean squared error.

## Data Availability

According to the internal regulations of the Institute, all data are the property of the Institute and can only be provided anonymously after an official request.

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
