# Peer review of "The Value of Left Ventricular Mechanical Dyssynchrony and Scar Burden in the Combined Assessment of Factors Associated with Cardiac Resynchronization Therapy Response in Patients with CRT-D"

_jcm, 2023, doi:10.3390/jcm12062120_

Round 1
Reviewer 1 Report
This is a single arm study of 56 patients with heart failure with reduced left ventricle Ejection Fraction (HFrEF) who underwent implantation of cardiac resynchronization therapy defibrillator (CRT-D), aimed at identifying possible predictors of response to CRT.
Enrolled patients had compensated NYHA II-III ischaemic or non-ischaemic HFrEF, with left ventricle Ejection Fraction (LVEF) < o =35%, QRS wave duration > o = 150ms with left bundle branch block (BBsx) morphology, and optimized medical therapy for at least 3 months, no recent infarction or indication for revascularization or heart transplantation.
Before implantation, patients underwent myocardial perfusion imaging and gated blood pool single photon emission computer tomography (SPECT) with calculation of the regional scar burden (the summed rest score, SRS), the regional entropy (E) and standard deviation (SD), the left and right ventricle peak filling rate 1 and 2 (LV and RV PFR1 and PFR2), the peak empting rate (PER), the interventricular dyssynchrony (IVD; calculated according to the histogram peak of LV and RV), transthoracic (TT) echocardiogram and 6 minute walking test. Then, 6 months after implantation, control TT echocardiogram was performed and patients were divided into two groups: patients who had an improvement of LVEF> o = 5% and/or areduction of End Systolic Volume (ESV) > o =15%, that were called CRT-responders and non-responders. In multivariate logistic regression on scintigraphic parameters, the left ventricle anterior wall standard deviation (LV AW_SD) was significatively higher in responders’ group, and the summed rest score (SRS) and prevalence of non ischemic etiology were significantly lower.
What makes this paper interesting is that it enters in a field where there are still a gap in the evidence. In fact, 30% of patients with indication to CRT according to the Guidelines do not obtain clinical benefit from it, suggesting that the existing criteria to CRT are not always effective.
However, while the standard TT echocardiographic parameters have not been shown to predict response, speckle traking echocardigoraphy and CMR parameters of heart dyssynchrony have already been demonstrated to predict CRT response (Ref. Comparison of strain imaging techniques in CRT candidates: CMR tagging, CMR feature tracking and speckle tracking echocardiography Wouter M. van Everdingen; Strain imaging to predict response to cardiac resynchronization therapy: a systematic comparison of strain parameters using multiple imaging techniques, Zweerink; Regional Strain Pattern Index—A Novel Technique to Predict CRT Response, MichaÅ‚ Orszulak), and there are also trial that have demonstrated the predictability of the cardiac scar burden (Ref. Value of left ventricle mechanical dyssynchrony and scar burden in combined assessment of factors associated with cardiac resynchronization therapy response Tariel A. Atabekov).
Anyway the study is designed in a linear way, the objectives are clearly expressed, however some formal improvements could be made especially in the introduction: we suggest quoting better references regarding the epidemiology of HF and the indication CRT, for example from the Guidelines, furthermore recent reviews about CRT non-response could also be cited (Ref. Prevention of non-response to cardiac resynchronization therapy: points to remember Huolan Zhu; Non-response to Cardiac Resynchronization Therapy Syed Yaseen Naqvi; Cardiac Resynchronization Therapy State-of-the-Art of Current Applications, Guidelines, Ongoing Trials, and Areas of Controversy Frits W. Prinzen). Moreover positive results of the already studied echocardiographic and MR parameters of dyssunchrony should be quoted. (Ref. Comparison of strain imaging techniques in CRT candidates: CMR tagging, CMR feature tracking and speckle tracking echocardiography Wouter M. van Everdingen; Strain imaging to predict response to cardiac resynchronization therapy: a systematic comparison of strain parameters using multiple imaging techniques, Zweerink; Regional Strain Pattern Index—A Novel Technique to Predict CRT Response, MichaÅ‚ Orszulak). Since a surrogate outcome of echocardiographic remodeling was used, its validity as an alternative to a clinical outcome must be supported by adeguate references (Ref. Ellenbogen KA, Gold MR, Meyer TE, Fernndez Lozano I, Mittal S,Waggoner AD, et al. Primary results from the SmartDelay determined AV optimization: a comparison to other AV delay methods used in cardiac resynchronization therapy (SMART-AV) trial: a randomized trial comparing empirical, echocardiography-guided, and algorithmic atrioventricular delay programming in cardiac resynchronization therapy. Circulation. 2010). Furthermore, we would also insert the legend with the abbreviations, and some graphs regarding how the MD parameters are obtained.
The patients enrolled in the study were well selected because they had class I recommendation for CRT implantation in the most recent European Guidelines on resynchronization and HF and American Guidelines on HF, even patients’ NYHA classes were the ones with strongest evidence (Ref. Comments on the 2021 ESC guidelines on cardiac pacing and cardiac resynchronization therapy, Rev Esp Cardiol. 2022), but I would extend the exclusion criteria to recent (within 3 months) revascularization, other cardiac intervention or acute decompensation.
Furthermore, as pointed out by the authors themselves, patients’ sample is small, follow up (FU) is brief, and enrollment is in a single center. From our point of view, considering the high prevalence of HFrEF, the wide indication to CRT-D implantation and the annual number of implants performed in large centres, the number of enrolled patients appears to be very small. This fact could suggest a selection bias. It also makes difficult to obtain significance of major clinical outcomes such as death and hospitalization which in fact are not considered in this trial. However, other minor clinical outcomes (e.g., NYHA class change and 6MWT) could be verified. Mention is made about the 6MWT performed before the implantation but this is not followed by any post-implantation data.
The two groups of patients were homogeneous in terms of demographic and clinical characteristics, but more importantly they were homogeneous in pre-implantation echocardiographic parameters, but this is never mentioned before the results paragraph and these parameters are not reported in the table of characteristics of population.
Implant success was similar in both groups as evidenced by the QRS length, the device setting was the same, and the BIV stimulation rate was high and consistent in both groups.
Furthermore, it would have been important to verify the homogeneity of the therapy, especially the beta-blocker in the two groups.
The positive response results could potentially be linked only to the possibility of titration of the therapy. And so a different indication to CRT or further one would arise.
Overall, the study is very interesting, althought the use of a score measuring the number of heart segments with scar is redundant when the main parameters seem to be related to the scar of the anterior wall and the free wall of the RV.
Author Response
Thanks for the review. Answers are in the attached file.

Reviewer 2 Report
Lines 160 and 290-300: please unify the abbreviations: PER - peak ejection/emptying rate.
Lines 271-274: please be more specific than "...ROC curved showed signifficant differences..."
Line 275: "years 2015-2020" instead of "2015-2020 years). Also please provide total number of patients that received CRT-D in this timeframe.
Author Response

(The authors gave the same response as above.)

Reviewer 3 Report
This study aimed to identify the significance of mechanical desynchrony and scar burden assessment using magnetic resonance imaging and gated blood pool SPECT as a combined predictor of cardiac resynchronization therapy response in patients with HF of ischemic and nonischemic etiology. Although this work has been conducted and clinically analyzed, however, the technical contribution of the paper needs to be reevaluated because of the following:
- The manuscript title of this article should be modified to include CRT-D.
- More details, such as type and specification regarding CRT-D, must be included.
- Authors need to add MRI images before and after to the study results.
- The sample size is small; the authors should explain how they obtained it through random or convenience sampling.
- The clinical criterion of the study needs more explanations with references.
- Sections 2.3 and 2.5 should be supported with references.
Author Response

(The authors gave the same response as above.)

Round 2
Reviewer 3 Report
The manuscript has been enhanced, thanks to the authors.